# Alterations in ^13^C and ^15^N Isotope Abundance as Potential Biomarkers for Tumor Biology and Risk Factors for Cervical Lymph Node Metastases in Oral Squamous Cell Carcinoma

**DOI:** 10.3390/cancers17183047

**Published:** 2025-09-18

**Authors:** Katarzyna Bogusiak, Piotr Paneth, Józef Kobos, Marcin Kozakiewicz

**Affiliations:** 1Department of Maxillofacial Surgery, Medical University of Lodz, 247/249 Pomorska, 92-209 Lodz, Poland; marcin.kozakiewicz@umed.lodz.pl; 2Institute of Applied Radiation Chemistry, Lodz University of Technology, 116 Żeromskiego, 90-924 Lodz, Poland; piotr.paneth@p.lodz.pl; 3Department of Histology and Embryology, Medical University of Lodz, 7/9 Żeligowskiego, 90-752 Lodz, Poland; jozef.kobos@umed.lodz.pl

**Keywords:** oral cavity cancer, stable isotope ratio mass spectrometry, prognostic indicator, tumor spread, metabolic reprogramming, biomarker development

## Abstract

This article presents a novel approach to assess cancer cell metabolism in oral squamous cell carcinoma (OSCC) using isotopic ratio mass spectrometry (IRMS). We investigated if IRMS-based isotopic profiling could reflect metabolic dysregulations associated with disease progression. This topic is both timely and clinically relevant. This is due to the fact that patients with OSCC have poor prognosis, which is related to lymph node involvement. In this prospective study, we analyzed tumors derived from 61 patients. We measured the relative abundance of carbon ^13^C and nitrogen ^15^N in the samples. Although these IRMS parameters were not independently predictive of lymph node status, they were associated with key adverse prognostic factors. We believe that IRMS could serve as a promising adjunctive biomarker and may complement classical histopathological evaluation.

## 1. Introduction

Isotope ratio mass spectrometry (IRMS) is an analytical method useful for measuring the relative abundance of selected isotopes. Lately, this technique has gained popularity and new applications in biomedicine [1,2,3,4,5]. Utilization of IRMS in this field of science is supported by the fact that various tissues of the human body have different isotopic compositions, as well as by the fact that isotopic abundance is influenced by the metabolic pathways in cells that constitute tissues and organs. In biomedicine, stable isotopes of light chemical elements are mainly selected for analyses, like hydrogen (H), carbon (C), nitrogen (N), oxygen (O), and sulfur (S) [6,7,8,9]. It has been proven that various diseases are accompanied by disturbances in cell metabolism, leading to changes in metabolic pathways and consequently to changes in their isotopic composition. For example, Taran et al. observed altered nitrogen and carbon signatures in Wilms’ tumors, while Tea et al. demonstrated metabolic reprogramming reflected in isotope abundance in breast cancer [10,11].

IRMS offers numerous possibilities for research in cancer biology, as it reflects changes in cell metabolic reactions. Metabolic reprogramming, which is believed to be a major hallmark of cancerogenesis, includes several well-defined changes in cancer cell pathways. These changes help provide essential substances and energy to meet altered anabolism and growth needs of cancer cells. This complex process includes upregulation of aerobic glycolysis, glutaminolysis, lipid metabolism, an increased pentose phosphate pathway and amino acid metabolism, as well as mitochondrial changes [12,13]. There are many factors affecting oncogene-driven metabolic changes in metabolism, including oncogenes, tumor suppressor genes, growth factors, and tumor–host cell interactions, as well as the conditions of the microenvironment, such as hypoxia and oxidative stress [13]. It has also been observed that the degree of clinical advancement (staging) of malignant tumors is associated with varying degrees of deviations in the metabolism of cancer cells. The biochemical processes that constitute metabolic reprogramming at particular stages of carcinogenesis occur at different speeds [14]. Recently, several studies that used stable isotope ratio assessment revealed that there are some significant implications between IRMS measurements and clinical findings, such as disease-free survival time (bladder cancer), tumor aggressiveness (Wilm’s tumors) and propensity to be invasive (breast cancer) [10,11,15].

Malignant neoplasms derived from epithelial tissues are characterized, amongst other features, by early cervical lymph node metastases. It has been proven that carcinoma spread to lymph nodes is one of the major factors affecting the outcome of treatment of patients with oral squamous cell carcinoma (OSCC), decreasing the 5-year survival rate. Metastasis to regional cervical lymph nodes is related to the deterioration of tumor control (this increases the risk of loco-regional recurrence and distant metastases) [16]. Despite the advancements in radiological diagnosis, the pre-surgical detection of lymph node metastases is characterized by a relatively high rate of false positive and false negative cases. Sensitivity in detecting lymph node involvement with the use of standard radiological imaging methods ranges from around 60% to 85% for computed tomography (CT) and to about 90% for magnetic resonance imaging (MRI) [17,18,19,20,21]. It has been estimated that about half of patients with oral cancer have metastases to lymph nodes unilaterally or bilaterally at the time of initial diagnosis. The exact epidemiology data varies among different countries and is related to many factors, i.e., primary tumor size [22]. It is also hypothesized that the presence of occult metastases to lymph nodes can be correlated with a decreased survival rate. It is estimated that in advanced stages (T3 and T4), the risk of occult lymph node spread can be higher than 20–30% [23,24].

A more accurate method for stratifying the risk of lymph node metastases could improve overall treatment outcomes in patients with oral cancer. In this study, we investigated if IRMS measurements could serve as a useful method in assessing the risk of lymph node spread. It can be assumed that alterations in tumor biology evaluated at the isotopic level can be a good predictor of lymph node metastases. The aim of this study was to verify if changes in the abundance of nitrogen ^15^N and carbon ^13^C isotopes in tumor tissues in patients with OSCC could serve us independent predictors of lymph node metastasis. In addition, the correlation between clinico-pathological risk factors for cervical lymph node metastases and IRMS parameters was analyzed.

## 2. Materials and Methods

### 2.1. Study Design and Patient Cohort

This prospective study included 61 consecutive adult patients with OSCC treated surgically in our department, fulfilling the inclusion criteria. The cohort consisted of 24 females and 37 males aged 43 to 92 years (mean age of 66.3 ± 9.4). The inclusion criteria were a primary diagnosis of OSCC (confirmed by histopathological examination) and localized tumor advancement enabling radical resection with clear surgical margins (R0 resection, defined as no microscopic residual tumor at the resection margin). The exclusion criteria included prior malignancy or irradiation of the head and neck region, previous chemotherapy or antibody therapy, and the presence of distant metastases. The nutritional status of all patients was assessed using body mass index (BMI) and a panel of laboratory tests, including total protein, albumin, alanine aminotransferase (ALT), aspartate aminotransferase (AST), transferrin, glucose, glycated hemoglobin (HbA1c), inorganic phosphate, calcium, and magnesium levels. Patients with malnutrition (BMI < 18.5) or a diagnosis of diabetes mellitus were excluded from the study. No participant followed specific dietary restrictions or exclusionary regimens. This study was approved by the Bioethics Committee (RNN/185/18/KE).

All patients enrolled into this study underwent standard therapeutic procedures (tumor resection, neck dissection) and adjuvant treatment, if necessary, as recommended by the National Comprehensive Cancer Network (NCCN) Clinical Practice Guidelines in Oncology (NCCN Guidelines) [25]. The following demographic and pathological information was collected: gender, age at the time of diagnosis, primary tumor site, pathomorphological stage (pTNM—pathologic TNM) and grade, and lymph node status. Subsequently, using the IRMS procedure, we obtained the information on the isotopic abundance of ^15^N and ^13^C of samples derived from oral carcinomas. The patients were divided into two groups. The first group constituted individuals without lymph node metastases—LNM (−). The second group included patients with nodal spread—LNM (+). This division was made according to the pathological stage. 

Additionally, the lymph node ratio (LNR) was calculated for each patient in the pN(+) group. This parameter was defined as the number of lymph nodes with metastases divided by the total number of dissected lymph nodes (LNR = number of metastatic lymph nodes/total dissected lymph nodes). 

Histopathological assessment was combined with IRMS to provide a comprehensive understanding of nodal spread.

### 2.2. Preparation of the Samples

During surgical procedures, four tissue samples (approximately 2 mm × 2 mm each) were extracted from each patient’s tumor. The collected specimens underwent specific processing routes for IRMS analysis and histopathological assessment. Two samples were immersed in formalin, embedded in paraffin, and evaluated histopathologically by an experienced pathologist (JK). The entire postoperative tumor and lymph node specimens underwent routine histopathological examination, assessing features such as depth of cancer infiltration (DOI, measured in millimeters), tumor thickness and diameter, bone invasion, surgical margins status (R status), angioinvasion, neuroinvasion, number and localization of lymph node metastases, and extranodal extension (ENE). TNM staging, following the American Joint Committee on Cancer 8th Edition, was used [26]. The remaining two samples were frozen at −70 °C for IRMS analysis.

### 2.3. IRMS Procedure

IRMS analysis of δ^15^N and δ^13^C isotopes was performed on 82 tissue samples from OSCCs. The samples were frozen at −70 °C for 48 h, lyophilized using a Christ Delta 1–24 LSC lyophilizer (GmbH, Osterode am Harz, Germany), and approximately 3 ± 1 mg of each sample was weighed into tin capsules for IRMS analysis. On average, three samples were obtained from one tissue section. Vanadium pentoxide served as the combustion catalyst, and thiobarbituric acid, calibrated against atmospheric nitrogen and Pee Dee Belemnite (PDB), was the primary reference standard for δ^15^N and δ^13^C, respectively.

To ensure reproducibility, repeated calibrations using certified reference materials were performed. In this study, standard IRMS validation protocols were followed. Calibration reproducibility was assessed through statistical evaluation of replicate measurements, including standard deviations and control chart analysis.

IRMS measurements were conducted using a Sercon SL20–22 Continuous Flow Isotope Ratio Mass Spectrometer connected to a Sercon SL elemental analyzer for simultaneous carbon–nitrogen analysis. Isotopic ratios were expressed as δ values, calculated using the following formula:δX(‰) = (R sample/R standard − 1) x 1000 (1)
where X represents δ^15^N or δ^13^C, and R is the isotope ratio of the heavier to lighter isotope (^15^N/^14^N or ^13^C/^12^C). For carbon isotopes, δ^13^C values were compared to the ^13^C/^12^C ratio in the PDB standard; for nitrogen isotopes, δ^15^N values were determined relative to the ^15^N/^14^N ratio of atmospheric nitrogen. IRMS measurements were validated using control samples (standard reference materials) with a standard deviation of ±0.2‰ for δ^13^C and ±0.3‰ for δ^15^N, ensuring repeatability and accuracy.

Additional IRMS parameters analyzed included the minimal (Min) and maximal (Max) percentage mass contents of carbon (C) and nitrogen (N), median and interquartile range (IQR), mean ± standard deviation (SD), percentage mass contents of C and N, and total nitrogen-to-carbon ratio ([N]/[C]).

### 2.4. Statistical Analysis

Statistical analyses were performed using Statistica, version 12.0 (StatSoft Inc., Tulsa, OK, USA). For univariate analysis of risk factors for cervical lymph node metastasis, the χ^2^ test of independence was applied. Williams’ correction, a statistical adjustment for χ^2^ tests in small sample sizes or multiple categories, was used to improve the accuracy of *p*-values [27]. Logistic regression was used for multivariate analysis to determine independent risk factors. One-way analysis of variance (ANOVA) was used to detect differences in IRMS measurements between clinical groups, with the Kruskal–Wallis test applied for non-normal data distributions (assessed via the Shapiro–Wilk test). Homogeneity of variance was checked with Levene’s test. A *p*-value < 0.05 was considered significant.

## 3. Results

### 3.1. Lymph Node Status (pN) and Neck Dissections

In total, 61 patients were enrolled in this prospective analysis. About 60% of the study group were men (*n* = 37) with a mean age of 64.2 ± 8.9 years. The rest of the group consisted of 24 women (39.4%) with a mean age of 69.5 ± 8.4 years.

Most patients (45.9%) were diagnosed with an advanced stage of the disease (pT4). Almost 80% of cases (*n* = 47) were staged as class IV according to the AJCC 8th edition. Most patients (60.7%) had a positive history of cigarette smoking, and 28.3% of alcohol abuse. Analysis of histopathological grading showed that the majority of tumors were intermediately differentiated (G2) in 61 patients. The frequencies of G1 and G3 diagnoses were quite similar in 10 of the G1 patients (16%). OSCCs were well differentiated (G1), and in 11 cases (18%), they were poorly differentiated (G3) (Table 1).

The majority of OSCCs involved the floor of the mouth (*n* = 18, 29.5%) and the lower gingiva (*n* = 18, 29.5%). Detailed data is presented in Figure 1.

Neck dissections were performed on the whole group of patients (61 cases) enrolled in the study. The extent of lymphadenectomy (the levels of cervical lymph nodes removed and one-sided or bilateral neck surgery) depended on the primary tumor site and preoperative radiological assessment of the neck lymph nodes. Overall, we performed 115 neck dissections. These included selective neck dissections (SNDs), specifically lymphadenectomies limited to levels I–III (supraomohioid neck dissections, SOHNDs) and surgeries limited to levels I–IV (extended supraomohioid neck dissections, ESOHNDs). Selective neck dissection was performed only for cN0 malignancies. Radical neck dissection was performed in cN3 cases, including bulky metastatic diseases by the accessory nerve and cases with multiple clinically metastatic lymph nodes. The remaining patients with cN(+) (where nodal spread did not involve the accessory nerve, internal jugular vein, and/or the sternocleidomastoid muscle) underwent modified radical neck dissections. SNDs prevailed in this study, with a total of 91 procedures performed. Levels I–V dissections, including radical neck dissections (RND) and modified radical neck dissections (MRND), were performed 24 times. (Table 2).

During all SND procedures, a total of 1450 cervical nodes were cleared (ranging from 9 to 33 lymph nodes), averaging 15.9 nodes per procedure. During level I–V dissections, a total of 572 nodes were resected (ranging from 14 to 42 lymph nodes per procedure), averaging 23.8 nodes per procedure. Taking into consideration all types of neck dissections, the median lymph node yield was 15 nodes per procedure (ranging from 9 to 49). In total, 2022 lymph nodes were dissected during all surgeries, averaging 33.2 nodes per patient.

In 33 out of 61 patients, cervical lymph node metastases were confirmed by histopathological examination. The total number of positive lymph nodes resected during all procedures was 123, with an average of 3.7 and median of 3.0 (ranging from 1 to 9). The median lymph node ratio (LNR) was 0.10690 (ranging from 0.02222 to 0.36). (Table 3).

Analysis of nodal involvement revealed that pN3b status was most frequent in patients with cervical lymph node metastases. This was determined in 28 cases (45.9%). A minority of this study group were patients in the pN1 stage (1 case), pN2b (3 cases), and pN2c (1 case). The rest of our cohort were patients without lymph node spread (*n* = 28).

Contralateral metastases were found in nine cases, while bilateral lymph nodes involvement were observed in eight patients. Metastases larger than 3 cm occurred in 17 patients. Single metastases were observed in 26 cases, while multiple metastases were observed in 8 individuals. Lymph node spread was most commonly present in patients with tumors involving the lower gingiva (*n* = 12, 36.4%) and the floor of the mouth (*n* = 11, 33.4%). In our cohort in patients with lower lip cancer, we did not observe regional lymph node involvement. The distribution of lymph metastases within the different primary tumor sites is presented in Figure 2.

Histopathological assessment revealed that in most patients in the pN(+) stage, extracapsular nodal spread was present. This was observed in 27 patients (79.4%).

Subsequently, a univariate analysis of risk factors of cervical lymph node metastasis was performed. The results indicate that factors like male gender (*p* < 0.05), age under 65 year old (*p* < 0.05), smoking (*p* < 0.05), stage IV of clinical advancement of oral cancer (assessed according to the 8th edition of the American Joint Committee on Cancer (AJCC)), (*p* < 0.0000), presence of angioinvasion and/or neuroinvasion (*p* < 0.05), DOI > 10 mm (*p* < 0.05), and presence of keratosis (*p* < 0.05) were statistically important risk factors of regional lymph node involvement (*p* < 0.05). Additionally, primary tumor site, as well as histopathological grading, were found to be related to cervical lymph node involvement. In these cases, due to small sample sizes, statistical analysis (χ^2^ test) was applied with Williams’ corrections. Metastases were most commonly observed in patients with tumors involving the lower gingiva (*n* = 18, 29.5%) and the floor of the mouth (*n* = 18, 22.5%), (*p* < 0.00000). Well-differentiated tumors (G1) in histopathological examination were related to negative lymph node findings (*p* < 0.00000). Other analyzed factors proved to have no impact on the neck lymph node status (*p* < 0.05). The detailed data concerning nodal status is presented in Table 4.

### 3.2. Multivariate Logistic Regression Analysis of the Risk Factors of Cervical Lymph Node Metastasis

The following statistical variables were included in the multivariate logistic regression analysis: age, gender, smoking status, depth of invasion, tumor stage, presence of angio/neuroinvasion, and keratosis. The age, gender, and tumor staging were found to be independent risk factors of cervical lymph node metastasis (*p* < 0.05). Age was identified as a protective factor (OR = 0.869, 95% CI: 0.78–0.97), suggesting that older age is associated with a reduced risk of lymph node metastasis. Female sex was also limited to a lower risk of pN(+) status compared to male sex (OR = 0.22, 95% CI: 0.049–0.997). Tumor stage remained the strongest prediction with very low odds ratios for stages I, II, and III compared to stage IV. The exact data is presented in Table 5.

### 3.3. IRMS Measurements of Nitrogen ^15^N and Carbon ^13^C in Tumor Tissues

The abundance of nitrogen ^15^N and carbon ^13^C were measured with the use of IRMS in samples derived from tumor tissues from both analyzed groups of patients—LNM (−) and LNM (+). The percentage contents of these isotopes are presented as the minimum, maximum, standard deviation, and median values.

No significant correlation was found between the isotopic values and lymph node metastases. Statistical analysis revealed that the nodal status did not affect the values of the mean percentage mass contents of nitrogen 15 and carbon ^13^C (*p* > 0.05). Similarly, in both compared groups (patients pN0 and pN(+)), other analyzed IRMS parameters did not differ significantly (*p* > 0.05). The detailed results are shown in Table 6.

### 3.4. Correlation Between IRMS Measurements and Risk Factors of Cervical Lymph Node Metastasis

A statistical analysis was performed to investigate the correlation between the isotope abundance of ^13^C and ^15^N in tumor tissues and factors that proved to be important risk factors of the cervical lymph node metastasis in this study. Most comparisons did not reach statistical significance (*p* > 0.05). However, patients with advanced clinical stage (AJCC stage IV) demonstrated significantly higher median nitrogen content (13%) compared to those in stages I–III (12%). (Table 7a) Additionally, a statistically significant difference in carbon isotopic composition (δ^13^C) was observed between these groups—patients in stage IV had lower δ^13^C values (median −22.40‰) compared to earlier-stage cases (median of −22.88‰; *p* < 0.05‰), indicating a potential shift in carbon metabolism in more advanced disease (Table 7b). Furthermore, patients with angioinvasion or neuroinvasion also showed significantly lower δ^13^C values (−22.26‰) than those without these negative prognostic factors (−22.75‰; *p* < 0.05‰).

## 4. Discussion

The results of this prospective study provide valuable insight into the potential utility of IRMS in understanding the biological background of oral cancer, especially regarding tumor progression and lymph node involvement. Although no statistically significant differences in the isotopic abundance of ^15^N and ^13^C were observed between patients with and without lymph node metastasis, we recognized noteworthy associations between IRMS parameters and clinicopathological features. Significant associations were noted between advanced clinical tumor stage and histopathological features, such as the presence of angioinvasion or neuroinvasion. Specifically, the average nitrogen ^15^N content was higher in patients with more advanced clinical stages (*p* < 0.05‰), and the median δ^13^C was lower in stage IV (−22.40‰) compared to stages I–III (−22.88‰) (*p* < 0.05‰). Patients with angioinvasion or neuroinvasion also exhibited a lower median δ^13^C (−22.26‰) compared to those without these features (−22.75‰) (*p* < 0.05‰). These observations align with the existing literature. For instance, the findings reported in the present study are consistent with our previous research, which demonstrated that the isotopic composition of OSCC tissues is associated with tumor aggressiveness and clinical advancement [28]. In our earlier study, we observed that the mean nitrogen content was significantly higher in patients with stage IV disease compared to those with stage II–III (11.89% vs. 11.12%; *p* = 0.04‰), while the δ^13^C values were significantly lower in more advanced tumors (−22.69‰ vs. −23.32‰; *p* = 0.04). These results are concordant with previous studies, confirming that higher nitrogen content correlates with more advanced disease, and that there is a negative correlation between δ^13^C and tumor progression. Moreover, the presence of angioinvasion and neuroinvasion was associated with altered isotopic abundance. Previously, we reported a non-significant trend toward lower δ^13^C in tumors exhibiting angioinvasion (−22.16‰ vs. −23.17‰). The depletion of δ^13^C can be related to enhanced glycolytic flux and lipogenesis, and δ15N enrichment to increased amino acid turnover and nucleotide synthesis. These observations support the notion that metabolic reprogramming reflected in isotopic shifts may underlie aggressive histological features.

In the above-mentioned article, we also demonstrated that IRMS measurements can distinguish OSCC tissues from margins and healthy tissues. Similarly, Madej et al. utilized the natural abundance of ^13^C in urothelium as a marker for monitoring patients with bladder cancer [15]. In pediatric oncology, Taran et al. investigated the isotopic composition of Wilms’ tumors, suggesting IRMS as a biomarker for individualized cancer treatment approaches [10].

Unlike our earlier cross-sectional analyses focusing on tissue–margin contrasts and site/stage stratification, the present prospective study emphasizes nodal risk, incorporates multivariable modeling of established clinical predictors, and examines δ^13^C/δ^15^N in relation to adverse features (angio-/neuroinvasion).

Our findings are consistent with the data presented in the literature, indicating that variations in isotopic ^13^C and ^15^N contents can differentiate between malignant and non-malignant tissues, as well as between aggressive and indolent tumor phenotypes [10,11,15].

These observations are consistent with the metabolic reprogramming that occurs in the cancer cells. Advanced tumors often exhibit enhanced glycolysis, glutaminolysis. It can be assumed that more aggressive tumor behavior can be related to dysregulation of carbon metabolism. Similarly, the pattern of the ^15^N isotope content suggests that upregulation of the turnover of nitrogen-rich biomolecules (such as amino acids and nucleotides) is related to tumor progression. 

Nonetheless, the global comparison of IRMS parameters between LNM(+) and LNM(−) groups failed to reach the level of statistical significance. Several explanations may account for this finding. First of all, it is possible that the metabolic profiles of primary oral tumors are similar regardless of their metastatic status. IRMS measurements capture the biochemical environment and nutrient molecules utilized by cancer cells. However, the ability to metastasize is driven by a distinct set of factors, such as genetic mutations, epithelial-to-mesenchymal transition, and tumor–stromal interactions, which may not be reflected in isotopic variations. Moreover, isotopic abundance was measured only in samples from primary tumors. The metabolic microenvironment of these tumors may differ from that present in metastases in lymph nodes. Additionally, the sample size, inter-patient variability, and biological heterogeneity may have been confounding factors that affected our results. The sample size of 61 patients may have been insufficient to detect subtle correlations between IRMS parameters and lymph node status. 

Our observations suggest that isotopic changes may serve as indicators of overall tumor biology related to metabolic reprogramming rather than direct predictors of lymph node metastasis.

From a clinical point of view, the relevance of our findings lies in their potential to improve the stratification of patients with poor prognosis and those at risk for lymph node metastases. Current radiological methods, such as computed tomography (CT) and magnetic resonance imaging (MRI), have limitations in detecting occult metastases, with sensitivities ranging from 60 to 90% [17,18]. In our cohort, approximately 54% of patients had histopathologically confirmed cervical lymph node metastases, which is consistent with the reported prevalence of nodal involvement in OSCC [22,24,29].

Our study also identified important risk factors for cervical lymph node metastases, namely male gender, age under 65 years, smoking, advanced clinical disease stage, angioinvasion/neuroinvasion, depth of infiltration (DOI > 10 mm), histopathological grade, and presence of keratosis. These findings are consistent with previous reports linking these factors to adverse outcomes in OSCC patients [16]. Additionally, primary tumor site also proved to be correlated with the risk of nodal spread. However, the small sample sizes of the subgroups limit the power of statistical analysis. Notably, the primary tumor site, particularly the floor of the mouth and the lower gingiva, was strongly associated with lymph node involvement (*p* < 0.00000), likely due to the rich lymphatic drainage in these regions. The absence of lymph node metastases in lower lip cancers in our cohort further supports the site-specific nature of metastatic risk, as lower lip tumors are known to have a lower propensity for nodal spread [22,30].

Nevertheless, further statistical analysis found only three independent risk factors for nodal metastases in our cohort, including younger age (<65 years), male gender, and advanced clinical stage. Importantly, we found that the isotopic parameters also varied with these factors. These observations support the hypothesis that the metabolic profile assessed by IRMS reflects the underlying tumor biology associated with metastatic potential. 

A potential limitation to the broader application of the IRMS method is the possible impact of confounding factors (e.g., diet, nutritional status, and systemic metabolic diseases) on stable isotope ratio measurements. Although detailed dietary data were not collected in this study, a review of clinical records and participant interviews indicated that none of the participants adhered to exclusionary diets or restrictive nutritional regimens. All individuals reported consuming a typical mixed diet without omitting major food groups. This suggests a relatively consistent nutritional background across the cohort, although some dietary variability cannot be ruled out.

The results of our study suggest that IRMS might not replace traditional histopathological examination in assessing lymph node metastasis risk. However, it may serve as a valuable tool that may complement diagnostic methods by providing additional information on tumor metabolism. This may be particularly useful in a multimodal approach to risk stratification. Incorporating IRMS analysis together with clinical, pathological, and genetic markers may offer a comprehensive strategy that could better predict treatment outcomes and guide therapeutic decisions. For instance, lower δ^13^C values in tumors with angioinvasion or neuroinvasion may indicate more aggressive tumor biology, potentially justifying more intensive treatment or closer monitoring. 

Our study has some limitations. First of all, the sample size consisted of 61 patients (LNM(+) = 33; LNM(−) = 28). An a priori power analysis indicates the study provides approximately 80% power (α = 0.05, two-tailed) to detect medium-to-large group differences in isotopic metrics (Cohen’s d ≈ 0.7) between lymph node strata; smaller effects would likely require a larger cohort. Secondly, potential confounding factors, such as diet and nutritional status, can affect the isotope ratio measurements, as discussed above. Lastly, in our study, tissue sampling was performed only on primary tumors, not on metastatic lymph nodes. Without measurements from metastatic lymph nodes, the usefulness of IRMS as a biomarker for nodal spread is limited.

In future investigations, longitudinal assessment of δ^13^C and δ^15^N in both primary tumors and metastatic lymph nodes could offer deeper insight into the dynamics of tumor progression. Such paired analyses could directly test nodal isotopic signatures. Including other biologically relevant isotopes (e.g., sulfur, oxygen) could provide a more comprehensive metabolic profile. Furthermore, integrating IRMS with other molecular, metabolic, and genetic biomarkers (e.g., GLUT1/LDHA/GLS1) may enhance its predictive value and clinical utility. Further research could include experiments using cancer cell lines with varying metastatic potential, along with appropriate normal controls, under isotopically defined culture conditions, which would allow compound-specific isotope analyses. Additionally, to investigate correlations between key metabolites and ^13^C- or ^15^N-abundance, the endometabolome or intracellular metabolome should be analyzed in cancer and normal cell lines. Gas chromatography–combustion–isotope ratio mass spectrometry (GC-C-IRMS) could be used to assess the isotope ratio of each amino acid. The use of isotopic analysis in cancer research is still in its early stages, but preliminary results from this study and others suggest it has the potential to enhance our understanding of tumor metabolism and support its clinical application in the future.

## 5. Conclusions

Our study demonstrates that the isotopic abundance of nitrogen ^15^N and carbon ^13^C in oral cancer tissues did not independently predict lymph node metastases but was correlated with adverse prognostic factors. This suggests that the metabolic characteristics reflected by stable isotope composition in tumor tissues may not affect the propensity of nodal involvement. The results underscore the complexity of tumor biology, indicating that while isotopic profiling provides valuable insights into cellular metabolism and nutrient utilization, it may not directly relate with metastatic potential. Direct evidence of metabolic pathway alterations was not established in this study. Further research incorporating additional molecular and metabolic markers may help to better understand the association tumor metabolism with the risk of lymph node involvement. Studies integrating IRMS measurements with targeted metabolic experiments, like glucose uptake assays, GC-MS-based metabolomics, or tracer-based metabolic flux analysis, are needed to better understand metabolic reprogramming in OSCC.

Although IRMS parameters of carbon ^13^C and nitrogen ^15^N were not independently predictive of lymph node status, they were associated with key adverse prognostic factors, indicating their potential as adjunctive biomarkers that may complement classical histopathological evaluation. 

## Figures and Tables

**Figure 1 cancers-17-03047-f001:**
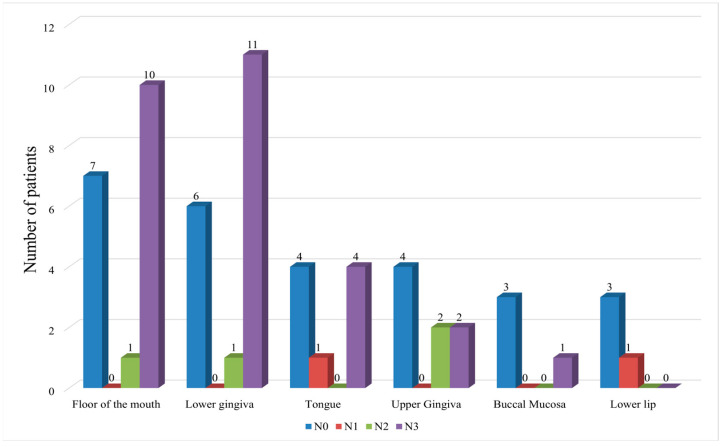
Distribution of OSCCs according to anatomical sites.

**Figure 2 cancers-17-03047-f002:**
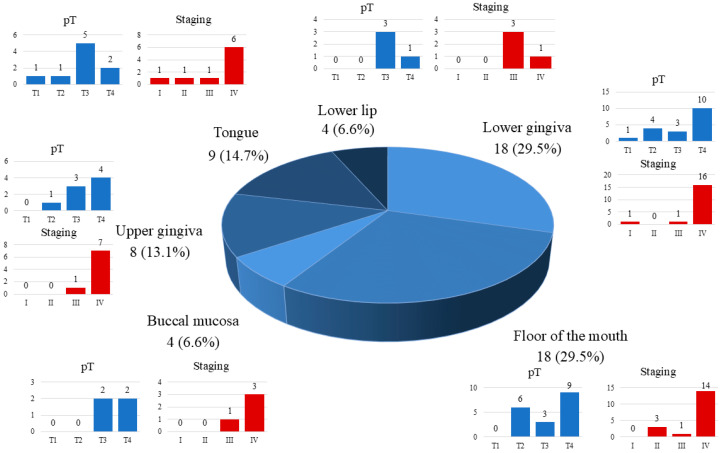
Distribution of lymph node metastases within the different primary tumor sites.

**Table 1 cancers-17-03047-t001:** Selected demographic and histopathological features in complete cohort (*n* = 61).

Characteristics	Number of Patients	
	*n*	%
Gender		
Male	37	60.7
Female	24	39.3
Age		
<65 years	23	37.7
≥65 years	38	62.3
Smoking		
yes	37	60.7
no	24	39.3
Alcohol consumption		
yes	17	27.9
no	44	72.1
pT stage		
T1	2	3.3
T2	12	19.7
T3	19	31.1
T4	28	45.9
pN stage		
N0	27	44.2
N1	2	3.3
N2	4	6.6
N3	28	45.9
Grading		
G1	11	18.0
G2	40	65.6
G3	10	16.4
AJCC stage		
I	2	3.3
II	4	6.6
III	8	13.1
IV	47	77.0

**Table 2 cancers-17-03047-t002:** TNM stages and types of neck dissection.

Number of Patients	cT	cN	pT	pN	Ipsilateral Side	Contralateral Side
2	1	0	1	0	SND	SND
1	1	0	2	+	SND	SND
1	1	+	2	+	MRND/RND	MRND/RND
4	2	0	2	0	SND	SND
2	2	0	3	0	SND	SND
2	2	0	3	+	SND	SND
1	2	0	2	+	SND	SND
1	2	0	3	+	SND	
5	2	+	2	+	MRND/RND	SND
2	2	+	3	+	MRND/RND	SND
5	3	0	3	0	SND	SND
4	3	0	3	+	SND	SND
1	3	0	3	0	SND	
1	3	0	4	0	SND	
1	3	0	4	+	SND	SND
1	3	0	4	+	SND	
1	3	+	3	+	MRND/RND	MRND/RND
1	3	+	3	+	MRND/RND	SND
1	3	+	4	+	MRND/RND	SND
9	4	0	4	0	SND	SND
3	4	0	4	0	SND	
3	4	0	4	+	SND	SND
7	4	+	4	+	MRND/RND	SND
2	4	+	4	+	MRND/RND	MRND/RND

**Table 3 cancers-17-03047-t003:** Characteristics of dissected lymph nodes in patients with cervical lymph node metastasis (per patient; *n* = 33 patients; 64 procedures).

	Mean	Median	SD	Min	Max
Lymph node yield	36.9	36	10.5	21	59
Number of positive lymph nodes	3.7	3	2.4	1	9
Lymph node ratio	0.10690	0.09090	0.07334	0.02222	0.36

**Table 4 cancers-17-03047-t004:** Characteristics and statistical analysis of clinical and pathomorphological features of analyzed group of patients, with and without lymph nodes metastases.

Characteristics	Number of Patients	Lymph Node Metastasis		χ^2^ Value	*p* Value
	n	Yes	No		
Gender				4.390	***p* < 0.05**
Male	37	24	13		
Female	24	9	15		
Age				5.838	***p* < 0.05**
<65 years	23	17	6		
≥65 years	38	16	22		
Smoking				4.390	***p* < 0.05**
Yes	37	24	13		
No	24	9	15		
Alcohol consumption				1.068	*p* > 0.05
Yes	17	11	6		
No	44	22	22		
Primary tumor site **				49.969 **	***p* < 0.05**
Buccal mucosa	4	1	3		
Floor of the mouth	18	11	7		
Lower lip	4	0	4		
Lower gingiva	18	12	6		
Upper gingiva	8	4	4		
Tongue	9	5	4		
Ulceration				0.236	*p* > 0.05
Yes	35	18	17		
No	26	15	11		
pT stage				0.100	*p* > 0.05
T1 + T2	14	8	6		
T3	19	11	8		
T4	28	14	14		
Depth of infiltration				4.573	***p* < 0.05**
≤10 mm *	28	11	17		
>10 mm	33	22	11		
Stage (AJCC 8th edition)				22.881	***p* < 0.05**
I + II + III	14	0	14		
IV	47	33	14		
Angioinvasion and/or neuroinvasion				6.573	***p* < 0.05**
Yes	26	19	7		
No	35	14	21		
ENE				N/A	N/A
Yes	27	27	0		
No	34	6	28		
Keratosis				6.416	***p* < 0.05**
Yes	53	32	21		
No	8	1	7		
Grade **				39.165	***p* < 0.05**
G1	11	4	7		
G2	40	24	16		
G3	10	5	5		

N/A—not applied. * The DOI was ≤5 mm only in 2 patients, and these were combined with patients with a DOI of 5–10 mm. ** χ^2^ value with Williams’ correction. The *p*-values below 0.05 are highlighted in bold to emphasize statistically significant results.

**Table 5 cancers-17-03047-t005:** Multivariate logistic regression analysis of the risk factors of the cervical lymph node metastasis in OSCC patients enrolled in this study.

Variables	B Value	S.E. Value	Wald χ2 Value	OR (95% CI)	*p* Value
Age	0.869	0.056	8.898	0.869 (0.779–0.970)	0.0029
Gender	0.220	0.770	4.133	0.22 (0.049–0.997)	0.0420
Tumor stage					
I	2.236	223.609	34.396	2.236 (1.030–4.854)	0.0000
II	1.379	158.117		1.379 (3.546–5.364)	
III	2.896	111.805		2.896 (1.865–4.263)	

S.E.—Standard error.

**Table 6 cancers-17-03047-t006:** Comparison of nitrogen ^15^N and carbon ^13^C abundance in tumor tissues in patients with lymph node involvement and without it.

		Lymph Node Metastasis	Lymph Node Metastasis	χ^2^ Value	*p* Value
		Yes	No		
Nitrogen (%)	Min–Max	3.10–13.40	6.40–13.10	0.195	*p* > 0.05
	Median	12.70	12.50		
	IQR	0.60	0.80		
Carbon (%)	Min–Max	44.00–69.50	44.10–63.90	0.103	*p* > 0.05
	Median	46.20	46.20		
	IQR	1.40	2.00		
[N]/[C]	Min–Max	0.045–0.316	0.104–0.291	0.641	*p* > 0.05
	Median	0.274	0.272		
	IQR	0.014	0.024		
δ^15^N(‰)	Min–Max	7.240–10.218	7.500–10.838	0.064	*p* > 0.05
	Median	8.800	8.700		
	IQR	1.117	0.945		
δ^13^C(‰)	Min–Max	−26.506–−20.072	−25.780–−20.858	0.103	*p* > 0.05
	Median	−22.304	−22.746		
	IQR	1.145	0.653		

**Table 7 cancers-17-03047-t007:** (**a**) Correlation between IRMS measurements and risk factors of cervical lymph node metastasis. (**b**) Correlation between IRMS measurements and risk factors of cervical lymph node metastasis.

(a)																									
Category		Nitrogen (%)												Carbon (%)											
		Min–Max			Median			IQR		χ^2^		*p* Value		Min–Max				Median		IQR		χ^2^		*p* Value	
Age																									
<65 years		3.10–13.30			12.0			2.10		0.15		*p* > 0.05		44.0–69.50				46.0		14.40		0.02		*p* > 0.05	
≥65 years		3.10–13.10			13.0			0.70						44.0–69.50				46.0		1.40					
Gender																									
Male		6.30–13.40			13.0			0.90		0.01		*p* > 0.05		44.0–63.90				46.0		2.60		0.04		*p* > 0.05	
Female		3.10–13.0			13.0			0.90						44.0–69.50				46.0		2.10					
Stage																									
I + II + III		6.40–13.40			12.0			6.10		4.12		*p* < 0.05		44.80–63.90				46.0		15.70		0.09		*p* > 0.05	
IV		3.10–13.40			13.0			0.90						44.0–69.50				46.0		1.70					
Smoking																									
Yes		3.10–13.40			13.0			0.90		0.03		*p* > 0.05		44.0–69.50				46.0		2.00		0.06		*p* > 0.05	
No		3.10–13.40			13.0			1.40						44.0–69.50				46.0		5.30					
DOI (mm)																									
≤10 mm		6.50–13.20			13.0			1.00		0.05		*p* > 0.05		44.0–63.90				46.0		5.40		0.08		*p* > 0.05	
>10 mm		3.10–13.40			13.0			0.80						44.0–69.50				46.0		1.60					
Angioinvasion or neuroinvasion																									
Yes		3.10–13.0			12.0			0.70		0.07		*p* > 0.05		44.0–69.50				46.0		1.60		0.03		*p* > 0.05	
No		5.50–13.40			13.0			1.40						44.0–69.50				46.0		3.60					
ENE (+) (ipsilateral or contralateral)																									
Yes		10.30–13.40			13.0			0.90		0.09		*p* > 0.05		44.0–50.80				46.0		1.40		0.12		*p* > 0.05	
No		3.10–13.40			12.0			1.10						44.0–69.50				46.0		3.60					
Keratosis																									
Yes		3.10–13.40			13.0			0.90		0.04		*p* > 0.05		44.0–69.50				46.0		1.60		0.05		*p* > 0.05	
No		6.60–13.40			12.0			5.20						45.80–63.70				46.0		13.70					
Grade																									
G1		6.40–13.10			13.0			6.20		0.06		*p* > 0.05		45.40–63.90				46.0		15.70		0.08		*p* > 0.05	
G2		3.10–13.40			13.0			0.90						44.0–69.50				46.0		1.70					
G3		3.10–12.70			12.0			9.40						45.80–69.50				46.0		23.70					
(**b**)																									
**Category**	**[N]/[C]**								**δ** ^15^ **N(‰)**								**δ** ^13^ **C(‰)**								
	**Min–Max**		**Median**	**IQR**		**χ^2^**	***p* Value**		**Min–Max**		**Median**		**IQR**		**χ^2^**	***p* Value**	**Min–Max**		**Median**		**IQR**		**χ^2^**		***p* Value**
Age																									
<65 years	0.045–0.316		0.27	0.066		0.08	*p* > 0.05		7.24–10.25		8.99		1.19		0.73	*p* > 0.05	−26.51–−20.07		−22.70		2.48		0.26		*p* > 0.05
≥65 years	0.045–0.286		0.27	0.009					7.74–10.84		8.65		1.34				−26.51–−20.86		−22.41		1.04				
Gender																									
Male	0.104–0.316		0.27	0.034		0.02	*p* > 0.05		7.24–10.25		8.65		1.01		0.12	*p* > 0.05	−24.69–−20.07		−22.44		1.52		0.03		*p* > 0.05
Female	0.045–0.291		0.27	0.010					8.19–10.22		8.83		1.31				−26.51–−21.29		−22.48		0.76				
Stage																									
I + II + III	0.104–0.291		0.27	0.168		0.07	*p* > 0.05		8.13–10.84		8.83 ± 0.87		1.02		0.11	*p* > 0.05	−24.68–−22.01		−22.88		1.68		4.25		*p* < 0.05
IV	0.045–0.316		0.27	0.013					7.24–10.22		8.86 ± 0.67		1.35				−26.51–−20.07		−22.40		2.48				
Smoking																									
Yes	0.045–0.316		0.27	0.012		0.04	*p* > 0.05		7.24–10.25		8.81 ± 0.76		1.33		0.09	*p* > 0.05	−26.51–−20.07		−22.45		1.44		0.02		*p* > 0.05
No	0.045–0.291		0.27	0.030					7.74–10.84		8.91 ± 0.64		1.35				−26.51–−20.86		−22.47		2.68				
DOI (mm)																									
≤10 mm	0.104–0.312		0.27	0.011		0.03	*p* > 0.05		7.24–10.84		8.81 ± 0.84		1.08		0.14	*p* > 0.05	−25.76–−20.07		−22.70		1.62		0.04		*p* > 0.05
>10 mm	0.045–0.316		0.27	0.010					7.74–9.86		8.88 ± 0.60		1.17				−26.51–−21.29		−22.70		1.52				
Angioinvasion or neuroinvasion																									
Yes	0.045–0.280		0.27	0.009		0.06	*p* > 0.05		7.90–10.22		8.92 ± 0.56		1.38		0.18	*p* > 0.05	−26.51–−20.07		−22.26		1.15		4.03		*p* < 0.05
No	0.045–0.316		0.27	0.034					7.24–10.84		8.80 ± 0.81		1.27				−26.51–−20.86		−22.75		2.68				
ENE(+) (ipsilateral or contralateral)																									
Yes	0.214–0.316		0.28	0.018		0.08	*p* > 0.05		7.24–10.22		8.87 ± 0.74		1.19		0.16	*p* > 0.05	−23.92–−20.07		−22.38		1.15		0.05		*p* > 0.05
No	0.045–0.291		0.27	0.026					7.74–10.84		8.83 ± 0.69		1.47				−26.51–−20.86		−22.71		2.68				
Keratosis																									
Yes	0.045–0.316		0.27	0.011		0.03	*p* > 0.05		7.24–10.22		8.84 ± 0.68		1.27		0.07	*p* > 0.05	−26.51–−20.07		−22.44		2.16		0.06		*p* > 0.05
No	0.104–0.291		0.27	0.150					7.50–10.84		8.89 ± 0.93		2.12				−24.69–−22.01		−22.72		2.02				
Grade																									
G1	0.104–0.312		0.27	0.168		0.05	*p* > 0.05		7.24–10.84		8.63 ± 0.95		1.09		0.13	*p* > 0.05	−24.68–−21.37		−22.70		1.83		0.07		*p* > 0.05
G2	0.045–0.316		0.27	0.014					7.50–10.22		8.90 ± 0.65		1.27				−26.51–−20.07		−22.41		2.48				
G3	0.045–0.278		0.27	0.229					8.55–9.66		8.89 ± 0.62		1.06				−26.51–−21.29		−22.58		4.21				

## Data Availability

The data on which this study is based will be made available upon request at https://www.researchgate.net/profile/Katarzyna-Bogusiak (accessed on 16 September 2025).

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
