# Peer review of "Alterations in ^13^C and ^15^N Isotope Abundance as Potential Biomarkers for Tumor Biology and Risk Factors for Cervical Lymph Node Metastases in Oral Squamous Cell Carcinoma"

_cancers, 2025, doi:10.3390/cancers17183047_

Round 1
Reviewer 1 Report
Comments and Suggestions for Authors
This manuscript presents an innovative application of IRMS in oral squamous cell carcinoma, exploring δ¹³C and δ¹⁵N isotopic abundance as potential biomarkers for tumor progression and nodal spread. While the study is well-conducted and clinically relevant, its conclusions are limited by modest sample size, lack of predictive power for lymph node metastases, and insufficient translational implications. The manuscript may be further improved by following suggestions.
- The word already mentioned in title should not be repeated in keyword. Select correct keyword relevant to the study.
- The study highlights novel use of IRMS in OSCC, but the sample size (n=61), 24 females and 37 males may be underpowered to detect subtle isotopic differences in relation to lymph node metastasis. A power calculation would strengthen the justification.
- The selection of δ¹³C and δ¹⁵N as the only isotopes may be too limited. Including other biologically relevant isotopes (e.g., sulfur, oxygen) could provide a more comprehensive metabolic profile.
- Although the paper claims novelty, previous studies by the same group and others have already reported similar findings. The incremental advancement over prior work should be made clearer.
- The clinical utility of IRMS remains unclear. Since δ¹³C and δ¹⁵N did not independently predict lymph node status, the translational value for clinical decision-making is limited.
- The study design is described as prospective, but the recruitment, follow-up duration, and outcome endpoints beyond histopathology (e.g., survival, recurrence) are not elaborated, reducing the impact of clinical correlations.
- Statistical analyses were applied broadly, but multiple testing corrections are not mentioned. Given the number of comparisons, false positives are possible.
- The study identifies associations with adverse prognostic features (e.g., angioinvasion, stage), yet does not sufficiently explain the biological mechanisms linking isotopic shifts to tumor aggressiveness.
- Tissue sampling was performed only on primary tumors, not on metastatic nodes. Without data from metastatic tissues, the interpretation of IRMS as a biomarker for nodal spread remains incomplete.
- Potential confounding factors such as diet, nutritional status, and systemic metabolic diseases (which influence stable isotope ratios) were not controlled or discussed.
- The discussion acknowledges limitations but does not provide concrete plans for integrating IRMS with other molecular or imaging biomarkers to improve predictive performance.
Comments on the Quality of English Language
May be improved.
Author Response
Ad 1. Addressed. Keywords revised to avoid duplication with the title and to better reflect scope.
Ad 2. We added a fragment to the discussion section.
Ad 3. We agree that including additional biologically relevant isotopes, such as sulfur, could indeed provide a more comprehensive view of the metabolic profile. In our current study, due to methodological constraints and the specific aims of this study, we limited our analysis to 13C and 15N. We focused on 13C and 15N also due to their well-established roles in tracing carbon and nitrogen fluxes in metabolic pathways. However, we acknowledge the potential of expanding the isotopic scope in future work, and we will consider incorporating sulfur and other relevant isotopes to enhance the depth of metabolic analysis.
Ad 4. Discussion fragment was added delineating novelty of our present versus previous studies.
Ad 5. Addressed. Tempered claims; positioned IRMS as adjunctive.
Ad 6. The present analysis focused on preoperative clinico-pathological endpoints, including pathologic nodal status and adverse histopathological features. Long-term outcomes (recurrence, disease-specific and overall survival) were not assessed in this dataset and are planned for a subsequent longitudinal report.
Ad 7. I appreciate your feedback. I would be grateful if you could elaborate on your comment regarding statistical analysis. Which statistical comparisons require my revisions.
Ad 8. We give a possible explanation in the discussion.
Ad 9. We added this comment to the discussion in limitations section
Ad 10. Regarding potential confounding factors affecting stable isotope ratios, we added a statement in the Methods noting that participants consumed a typical mixed diet and we included information on patients’ nutritional status. We also added a brief discussion of this issue.
Ad 11. We added information to the discussion.
Reviewer 2 Report
Comments and Suggestions for Authors
This manuscript explores the potential of Isotope Ratio Mass Spectrometry (IRMS) in evaluating the metabolic alterations in oral squamous cell carcinoma (OSCC) and its association with cervical lymph node metastases. A prospective cohort of 61 patients was analyzed, measuring the abundance of ¹³C and ¹⁵N in tumor tissues and correlating these with clinicopathological parameters. The study found no direct independent predictive value of these isotopes for lymph node metastases; however, correlations with adverse prognostic features (e.g., stage IV disease, angioinvasion/neuroinvasion) were observed. The authors suggest that IRMS may serve as an adjunctive biomarker to complement histopathological evaluation.
Major comments:
- Graphical abstract is not provided. Its inclusion could enhance clarity for readers.
- The title is informative but lengthy and could be made more concise without losing specificity. Avoid splitting words with line breaks (e.g., “bi-omarker”).
- The abstract contains a general flow but lacks explicit numerical data for key results (e.g., provide exact δ¹³C and δ¹⁵N ranges with p-values). Clarify that IRMS did not independently predict lymph node metastases but correlated with adverse prognostic factors.
- In the keyword section, consider including “stable isotopes” and “biomarker” for better indexing.
- The rationale is clear but somewhat lengthy. The funneling from general background to specific research gap can be more concise.
- The authors should explicitly state the hypothesis in the last paragraph: whether isotopic changes could serve as independent predictors of lymph node metastasis.
- Patient inclusion/exclusion criteria are described; however, the sample size (n=61) may limit statistical power. The authors should justify this.
- Isotopic signatures from clinical samples might be influenced by non-controlled epidemiological factors such as patient age. To strengthen the mechanistic interpretation, I recommend that the authors include experiments using cancer cell lines with varying metastatic potential, along with appropriate normal controls, under isotopically defined culture conditions, which would allow compound-specific isotope analyses.
- The endometabolome or intracellular metabolome should be analysed in cancer and normal cell lines to understand the correlations between key metabolites and 13C- or 15N-abundance.
- The authors should conduct 15N-abundance and amino acid content using gas chromatography - combustion - isotope ratio mass spectrometry (GC-C-IRMS) to understand the isotope ratios of each amino acid.
- I recommend that the authors evaluate 13C and 15N isotope signatures of saliva and plasma to predict nodal involvement, thereby increasing the translational relevance of the study.
- Details on tissue sample preparation are thorough, but the narrative on IRMS validation part could include more on calibration reproducibility.
- No significant correlation found between isotopic values and lymph node metastases. This should be clearly presented in the first lines of the Results to manage expectations.
- Figures lack detailed legends (e.g., axes labels in Fig. 1 & 2).
- Discussion appropriately contextualizes findings but tends to repeat results. Rather, the authors should appropriately provide a comparison of the results obtained in this study and similar studies.
- The limitations section acknowledges sample size but does not elaborate on potential selection bias or confounding factors.
- Figures lack sufficient schematic representation of the workflow. The authors should include one figure showing patient selection, sample preparation, and analysis steps.
- Some tables (e.g., Table 5 & 6) are dense and could benefit from visual summaries (heatmaps or simplified charts).
Minor comments:
- Expand abbreviations at first use (e.g., IRMS).
- Organize the methods to match the results section flow.
- Specify software version for statistical analysis in a consistent format.
- Use consistent decimal places for median/IQR reporting.
- Ensure uniform MDPI style (spacing, punctuation, DOI formatting).
- Check for typographical errors (e.g., “metas-tasis,” “bi-omarker”).
- Maintain consistent use of units (e.g., ‰ for isotopic ratios).
- Certain sentences are overly long and complex. Break into shorter sentences for clarity.
Author Response
Ad 1. Graphical abstract is provided.
Ad 2. We would prefer not to change the title. We corrected splitting words.
Ad 3. Numerical data is provided in the abstract.
Ad 4. It’s done.
Ad 5. We thank the reviewer for this remark. We agree that the Introduction is relatively detailed; however, we intentionally chose this structure to provide sufficient background for readers who may not be familiar with the use of IRMS in oncology. Since our work addresses a novel application of isotopic profiling in OSCC, we considered it important to first outline the broader biomedical relevance of stable isotope analysis and the clinical challenges in nodal staging. This broader rationale helps position the research gap more clearly for the readership of Cancers, which is interdisciplinary. For this reason, we would prefer to retain the current form of the Introduction, as it supports accessibility and contextual understanding of our study.
Ad 6. It’s done.
Ad 7. The paragraph was added to limitations section
Ad 8. We thank the reviewer for this insightful comment. We fully agree that experiments in well-controlled in vitro conditions (e.g., cancer cell lines of varying metastatic potential) would provide valuable mechanistic insights and allow disentangling clinical confounders such as patient age. However, the present study was designed as a prospective clinical cohort analysis, and cell line experiments were beyond its scope.
We have revised the Discussion to explicitly acknowledge this limitation and to highlight that future research combining IRMS in clinical samples with isotopically defined in vitro models would be essential to establish causal links between isotopic shifts and tumor cell metabolism.
Ad 9. We appreciate this valuable suggestion. We agree that parallel metabolomic profiling would provide deeper insight into which specific metabolic pathways underlie the observed isotopic shifts. Unfortunately, comprehensive endometabolome analyses were not feasible within the scope of the current clinical study.
To address this point, we now emphasize in the Conclusions that integration of IRMS with targeted metabolomic profiling in future work is an important next step to validate our interpretation of metabolic reprogramming in OSCC.
Ad 10. We thank the reviewer for this technically important suggestion. Indeed, GC-C-IRMS could resolve compound-specific isotopic compositions and would help to determine which amino acids are most affected by tumor metabolic alterations. While this approach was not available in our current setting, we acknowledge its potential for mechanistic insight.
We have added a statement in the Discussion highlighting the importance of compound-specific isotope analysis (CSIA) using GC-C-IRMS as a promising methodology for future studies.
Ad 11. We agree with the reviewer that non-invasive matrices such as saliva and plasma would significantly enhance the translational relevance of IRMS-based biomarkers. In our current project, only tumor tissue was available for isotopic analysis. However, we recognize the strong potential of body fluid analysis for risk stratification in OSCC and have mentioned this explicitly in the Discussion as a direction for future research.
We believe that integration of saliva and plasma isotopic profiling with tumor tissue analysis may ultimately help to establish clinically feasible biomarkers for nodal prediction.
Ad 12. Information on IRMS validation was added.
Ad 13. It is stated in the first lines of the section 3.3 of the results.
Ad 14. It’s done.
Ad 15. We haven’t found appropriate articles for comparison our results.
Ad 16. The limitations section was augumented.
Ad 17. We thank the reviewer for this suggestion. We agree that schematic workflow figures can be useful in some contexts. However, in our manuscript all steps of patient selection, tissue sampling, and IRMS analysis are already described in detail in the Materials and Methods section. Given the straightforward design of this study, we believe that an additional schematic figure would duplicate information already clearly presented in the text, without adding substantial value for the reader. For this reason, we would prefer to retain the current format without introducing a new workflow diagram.
Ad 18. We appreciate the reviewer’s suggestion to present Table 5 and Table 6 data using visual formats such as heatmaps or simplified charts. While we agree that graphical summaries can be informative, in this case we consider the tabular presentation more appropriate, as it allows readers to see the exact numerical values and statistical details that are essential for accurate interpretation. We are concerned that replacing or supplementing these tables with visual summaries might obscure important details. For this reason, we would prefer to keep the current format of tables as it best reflects the precision of our findings.
Ad ‘minor comments’. Suggested minor revisions have been implemented.
Reviewer 3 Report
Comments and Suggestions for Authors
The study is interesting and innovative. The objectives and results are clear and well documented.
-In section 2.1 of Materials and Methods (line 118-119) and in section 3.1 of Results (line 193) it is reported how all 61 patients underwent neck dissection. In line 216 it is reported how 33/61 patients had anatomopathological confirmation of lymph node metastasis and in table 3 pathological staging is reported (pT1+T2:14 - pT3:19- pT4:28). However, it is never indicated what the clinical TNM was of the 61 patients included in the study and within of the T parameter how many as T1, T2, T3, T4 and what the lymph node status of the patients was (N0 vs N+). doing so could justify the choice to carry out a neck dissection over the entire sample. It may be helpful to add a table with the patients' cTNM and pTNM and type of neck dissection performed to for further clarity it.
-In line 124-125-126 it is described how the patients were divided into 2 categories (LNM - and LNM+) was this division made according to the clinical or pathological stage?
-in line 206-207 it is reported as median lymph node yield wa 15 nodes per procedures (range from 9 to 49). Were the neck dissection with 9 lymph nodes considered adequate?
Author Response
Ad section 2.1.
We attached new table (Table 2) presenting cTNM and pTNM status and type of nec dissection
Ad line 124-126.
This division was made according to the pathological stage
Ad line 206-207.
We acknowledge that lymph node yield is an important marker for the adequacy of neck dissection and has prognostic implications in head and neck squamous carcinoma. While a yield of 9 lymph nodes is at the lower end of our median lymph node yield it is still within the acceptable range reported in the literature. A recent systemic review disclosed that minimum lymph node yield associated with improved survival outcomes varied from 10 to 36.5 dissected nodes, depending on tumour subsite and nodal status. However, establishing an universal cutoff is difficult, because many factors can influence the number of lymph nodes resected during neck dissections, like variability in surgical technique, anatomical subsite and pathological processing.
Importantly, all dissections in our study were performer by experienced head and neck surgeons, following oncologic principles. There was only one case with 9 lymph nodes resected with pN0 status. Therefore, we consider that it was oncological adequate.
Reviewer 4 Report
Comments and Suggestions for Authors
In this manuscript, Bogusiak et.al presents the use of IRMS in exploring the biology of oral cancer, indicating IRMS is good to show association between 13C, 15N incorporation and tumor progression, pathological features. Overall, the clinical data is detailed and well listed. But the authors need to improve the interpretation of the cancer biology using IRMS data and more biological experiment.
Major concerns:
- Line 326, “higher nitrogen content correlates with more advanced disease, and secondly that there is a negative correlation between δ13C and tumor progression.”. Can the authors use graphs to clearly show the positive or negative correlation? And what kind of correlation algorithm was used here?
- For the conclusion of metabolic reprogramming in OSCC cancer, the authors should also introduce some metabolism experiments, which can be corresponded with IRMS data. Then the conclusion will be more solid and convincing.
Author Response
Ad 1. The correlations between nitrogen content, δ¹³C values, and tumor stage have already been presented in detail in Table 7b. Since the data did not meet the assumption of normality (confirmed by the Shapiro–Wilk test), we applied Spearman’s rank correlation to evaluate the association.
Ad 2. We appreciate the reviewer’s insightful comment. We fully agree that functional metabolic experiments would provide important complementary evidence and strengthen the mechanistic interpretation of isotopic shifts. To address this point, we have revised the Conclusions
Round 2
Reviewer 1 Report
Comments and Suggestions for Authors
Author has significantly improved the paper, it can be accepted.
Reviewer 3 Report
Comments and Suggestions for Authors
The changes made to the text and the integration of the table are exhaustive and the answers to the questions are well argued and valid. It is a very interesting and well written paper.
Reviewer 4 Report
Comments and Suggestions for Authors
The authors have addressed all my questions. Thanks!